# Smart Android Based Home Automation System Using Internet of Things (IoT)

Muhammad Abbas Khan [1], Ijaz Ahmad [2],*, Anis Nurashikin Nordin [3], A. El-Sayed Ahmed [4],
Hiren Mewada [5], Yousef Ibrahim Daradkeh [6], Saim Rasheed [7], Elsayed Tag Eldin [8] and
Muhammad Shafiq [9],*

1   Department of Electrical Engineering, Balochistan University of Information Technology, Engineering and Management Sciences, Quetta 87300, Pakistan
2   Shenzhen College of Advanced Technology, University of Chinese Academy Sciences, Shenzhen 518055, China
3   Department of Electrical & Computer Engineering, International Islamic University Malaysia, Kuala Lumpur 43200, Malaysia
4   Mathematics Department, Faculty of Science, Taif University, P.O. Box 11099, Taif 21944, Saudi Arabia
5   Electrical Engineering Department, Prince Mohammad Bin Fahd University, Al Khobar 34754, Saudi Arabia
6   Department of Computer Engineering and Networks, College of Engineering in Wadi Alddawasir, Prince Sattam Bin Abdulaziz University, Al-Kharj 16278, Saudi Arabia
7   Department of Information Technology, Faculty of Computing and IT, King Abdulaziz University, Jeddah 22254, Saudi Arabia
8   Electrical Engineering Department, Faculty of Engineering & Technology, Future University in Egypt, New Cairo 11845, Egypt
9   Department of Information and Communication Engineering, Yeungnam University, Gyeongsan 38541, Korea
*   Correspondence: ijaz@siat.ac.cn (I.A.); shafiq@ynu.ac.kr (M.S.)

**Abstract:** Recently, home automation system has getting significant attention because of the fast and advanced technology, making daily living more convenient. Almost everything has been digitalized and automated. The development of home automation will become easier and more popular because of the use of the Internet of Things (IoT). This paper described various interconnection systems of actuators, sensors to enable multiple home automation implementations. The system is known as HAS (Home automation system). It operates by connecting the robust Application Programming Interface (API), which is the key to a universal communication method. The HAS used devices, often implemented the actuators or sensors that have an upwards communication network followed by HAS (API). Most of the devices of the HAS (home automation system) used Raspberry Pi boards and ESP8285 chips. A smartphone application has been developed that allows users to control a wide range of home appliances and sensors from their smartphones. The application is user-friendly, adaptable, and beneficial for consumers and disabled people. It has the potential to be further extended via the use of various devices. The main objectives of this work are to make our home automation system, more secure and intelligent. HAS is a highly effective and efficient computational system that may be enhanced with a variety of devices and add-ons.

**Keywords:** home automation systems; internet of things; application-programming interface; discipline

## 1. Introduction

Recently, Internet of Things (IoT) has been capturing the whole world. It is a network of connected devices that can be remotely monitored and controlled through the internet. IoT idea has been significant development in recent years [1]. It is now being utilized in various sectors, including smart homes, cloud computing (thread detection) [2–5], and heath care (biomedical disease) [6–9], telemedicine, industrial settings, and other applications. Incorporated into the IoT, wireless sensor network technologies allow worldwide connected smart devices with enhanced functionality. The wireless smart home automation system of the network consists of sensors and actuators that can share resources or communicate

with each other, which is the most important technology for creating smart houses. A "smart house" is a concept that is part of the IoT paradigm that aspires to incorporate home automation [10]. It is a significant advancement to enable consumers remotely monitor and control home appliances by connecting them to the internet. Several smart devices are available, including light switches that can be controlled with a voice command of a smartphone [11,12]. A smart irrigation system [13] and thermostats that can change interior temperatures while producing energy reports, reducing water waste, etc. Throughout the past several years, smart home solutions have been more popular [13]. Figure 1 presents a smart home automation system using several IoT-connected appliances.

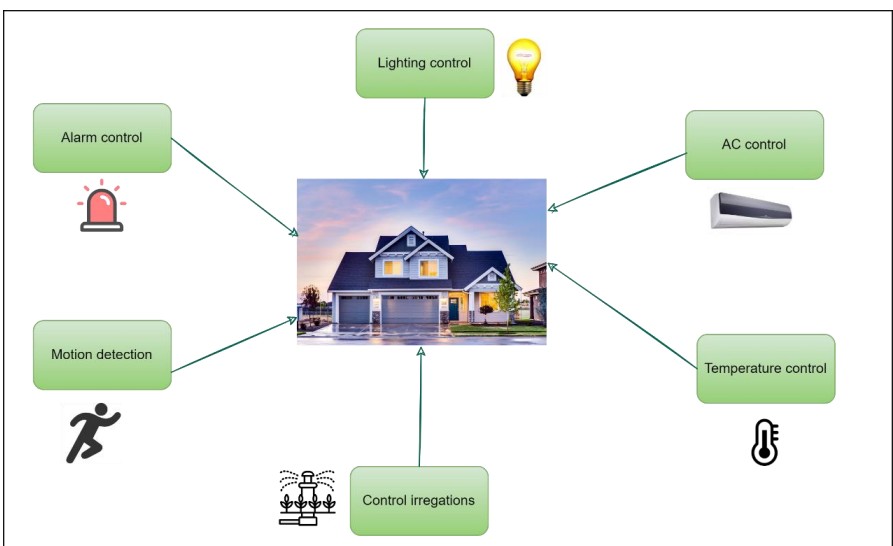

**Figure 1.** Shows smart IoT-based home automation system using various sensing devices.

Home automation systems are a technical method of control, feedback, actions of house appliances, and intelligent monitoring based on the demands of the home inhabitants. In recent times, most switching activities are manual and do not utilize the concept of IoT. A controlling device is very helpful to the user because it allows for managing and handling all appliances linked to the system from a single location. Home appliances such as fans, lights, and switches may be controlled remotely via a central control board [14].

One of the most significant benefits of home automation systems is the easy control and management of various devices such as desktops, laptops, smartphones, smartwatches, etc. The benefits of a smart home automation system are easy to manage and control all the home appliances including lighting control, temperature adjustment, Tv and air condition control, for safety purposes use video camera, etc. Which saves a lot of time and money.

Various researchers have proposed an IoT-based home automation system in the last two decades. Different technologies have been implemented in home automation systems; each has advantages and disadvantages. For instance, a home automation system based on Bluetooth [15,16] is inexpensive, fast, and simple to install, but it can only communicate for short distances. ZigBee and GSM (Global System for Mobile communication) are wireless technologies that work for long-distance communication. Zigbee [16,17] is a wireless mesh network standard for battery-powered devices in wireless control and monitoring applications. It is designed to have a low cost and low power consumption. Furthermore, it has a slow data speed, transmission, and low network stability.

Additionally, it has a high cost of maintenance. Wi-Fi-based technology is implemented in the literature [17–24]. Wi-Fi technology has several advantages over ZigBee and Z-Wave, including cost, complexity (which can be interpreted as simplicity), and accessibility.

Furthermore, smart devices that are enabled with Wi-Fi are typically affordable. Additionally, Wi-Fi is the cheaper option in which devices are easy to connect. After that, because Wi-Fi is now considered a key necessity and is installed in the majority of homes,

it is much simpler to acquire devices that have Wi-Fi capabilities built-in. Wi-Fi is very simple by using and connecting, which means that a user only needs to connect a limited number of devices for a home automation setup. This makes Wi-Fi ideal for home use. It does not any additional hardware for connecting the network. The basic configuration is required of a user to implement a smart HAS.

Additionally, Wi-Fi did not design for creating the mesh network. It uses six times much power as comparable devices implement Z-Wave, ZigBee, or Bluetooth, because most of the Wi-Fi routers might possible to connect twenty to twenty-five devices at the same time. Furthermore, Wi-Fi offers multi- benefits over Ethernet, for example simple connection and accessibility of different devices, the expandability (which allows for adding new devices have no burden of extra cabling), cost cheaper, and the necessity that just a single access point be present. The issue might prevent the Wi-Fi signal from being sent and cause problems for the devices that are connected. In addition, there are difficulties with speed of the connection (high speed Wi-Fi of the network is substantially slower than the speed of a wired network), as well as worries over Internet security and privacy. For the purpose of the open-source home automation hardware components and low-cost, such as Raspberry Pi and Arduino microcontroller unit (MCU) boards, as well as different sensors have been used. In [12,25–33], many home automation methods with Arduino boards are discussed. Arduino is an open-source hardware platform that is simple to use, inexpensive, highly flexible, and easy to program [34].

Additionally, large society is a significant strength of this product. On the other hand, Arduino was not intended to deal with the considerable complexity of more sophisticated projects. Raspberry Pi (RPi) has robust solution for complex applications requiring real-time processing. RPi is an exciting new technology available for a price significantly lower than any mobile device [35].

Most projects implemented open-source Raspberry Pi for online users. These communities are always excited about new projects that are being developed. Python is the preferred language for developing software for the Raspberry Pi because, compared to other programming languages, it is easier to use (requiring fewer lines of code and having a lower level of complexity). Raspberry Pi is inexpensive, has a low environmental impact, and does not need any cooling system. [36–40] present the ideas of using Raspberry Pi in conjunction with intelligent home automation. The inexpensive Wi-Fi modules known as ESP8266 chips are an ideal choice for use in Internet of Things (IoT)-related projects. The ESP8266 has a single processing core that operates at 80 megahertz [37–44] examples of home automation projects in which ESP8266 chips were employed. Table 1 compared different features of smart HAS existing in the previous literatures of the last ten years.

**Table 1.** Shows a comparison of proposed approaches, features relevant to home automation systems with articles published in the last 10 years.

| References | Controller | User Interface | Communication | Application |
|---|---|---|---|---|
| [16] | PIC (Microchip Technology in Chandler) | mobile application | Bluetooth | Manage the operation of indoor appliances |
| [7] | Arduino | mobile application | Bluetooth | Both indoor and outdoor control appliances at a short distance |
| [18] | Arduino mega | mobile application | Ethernet, ZigBee | Control indoor appliances |
| [19] | Raspberry PI | | ZigBee, Wi-Fi | Regulating humidity, temperature, brightness, movement, and electrical currents |
| [20] | PC server/Laptop | mobile application | ZigBee | Management of interior appliances; however, it is not put into action |
| [21] | TI-CC3200 | mobile application | Wi-Fi | Manage the operation of indoor appliances, and monitor the moisture of the soil |
| Proposed System | Raspberry PI, Arduino mega | Web-based and mobile applications | Wi-Fi | Multiple home automation (indoor and outdoor), security management, energy prediction, Statics power and management, and Google Assistant compatibility. |

Other home automation systems have commercial tools or platforms, such as Domintell, Qivicon, HomeSeer, Loxone or. This type of system falls under the category of "commercial platforms." They provide a wide selection of home automation devices from several different manufacturers as well as multiple automation including a locking system, temperature control, lighting system, environmental system, HomeSeer, Qivicon video surveillance, and the anti-intrusion features. Every solution includes a mobile application that can use to operate the systems. Regarding the cost, everything relies on the home's square footage and the user requirement. It is estimated that the bare minimum will cost between 2000 and 2200 dollars [44].

A wide selection of open-source HAS is available [43–50]. Home Assistant [46] and OpenHAB [43] are the most powerful participants in the community of open-source HAS. Both of these projects have a similar aim and integrate a large number of devices. On the other hand, to integrate devices, using openHAB needs knowledge of input commands; the process is complicated and time-intensive. On the other hand, Home Assistant is easier to use but needs substantial work to configure. Mobile applications seem to have less flexibility and appear fairly difficult and sophisticated, particularly for beginners. In the research of Domoticz [42] has presented a sufficient number of functions; most of its setting is carried out through web and plugin are implemented to increase the software's capability. The interface of the application is not easy to understand based on setup information which is Domoticz's proposed study limitations. Two French companies that are major participants in the open-source home automation ecosystem are Calaos [46] and Jeedom [47]. Unfortunately, most of the forums and communities are in French, which might be a language's barrier to the adoption in other parts of the globe. Table 2 provides a comparative analysis of the most important open-source home automation systems in their feature sets.

**Table 2.** Shows a comparative study of the most important and relevant home automation platforms based on their functionality.

| System | API | Programming Language | Other Possible Feature |
|---|---|---|---|
| [43] Open Hub | Representational state transfer (REST) | JAVA | Extensive documentation, many protocols, web interface, MQTT (Message Queuing Telemetry Transport |
| [44] Domoticz | Based on JSON (JavaScript Object Notation) | C++ | Few protocols, MQTT (Message Queuing Telemetry Transport), web interface, extensive documentation |
| [47] Jeedom | HTTP and JSON (JavaScript Object Notation) based | PHP | Extensive documentation, many plugins, a web interface, and many protocols. |
| [44] Home Assistant | Python/REST | Python | Extensive documentation, many protocols, web interface, MQTT (Message Queuing Telemetry Transport). |
| Proposed system | Rest and JSON based | Python | HTTP, web interface, expanded documentation (French, English, Chinese), few plugins, many protocols, Apache 2.0. |

The platforms may be distinguished from one another in several ways, including the programming language used for development, the API, the number of plugins and protocols are developed, and the quantity and variety of documentation. The writers of the article [46] provide a comprehensive analysis of fifteen different platforms of open-source.

The objective of this research is to represent HAS; this system has been designed and developed for multiple building/home automation such as security and access control [47], appliances control including thermostats, lights, AC, etc.), energy and power management [48,49].

This is very applicable in sustainability fields such as Environmental Sustainability (energy management technologies) [50], Economic sustainability (Cost-effective smart home design technologies) [51], Sustainability from smart homes automation system to smart space [52–56]. The ideas presented in this study are the extended version of the research [57]. The automatic building system has been proposed in this research to minimize the spreading and spreading of the COVID-19 under the pandemic working space by preventing individuals from touching particular surfaces and objects. Additionally, this solution was intended

to assist the buildings managing under the emergency condition. The primary goal of this research is to focus on smart home appliances, not on the COVID-19 situation.

Table 1 presents the research gap of the proposed study. Various factors include technologies, communication type, the controllers, and most importantly, the application for intelligent home automation solutions.

Many research articles presented different communication technologies, such as using wireless or implemented wireless communication technology to send data from nodes to storage devices, connect the sensor with nodes, etc. The HAS set up, Wi-Fi or Ethernet is enough. Most IoT devices support Wi-Fi. The best technology selection, such as a processer (controller) for a home automation system, depends on the necessities of the home by user wants. Most of the literature preferred to use the Raspberry family because of its strong computing abilities that make powerful than Arduino boards. Furthermore, we selected the Raspberry Pi board and the microcontroller system (ESP8266 chip) in the proposed study because this chip is small in size, has low power consumption, and has robust storage and processing. HAS builds powerful and flexible API use for all the devices to work combined. Further, HAS provides JavaScript data format.

Controlling programmable devices with a Transmission Control Protocol/Internet Protocol (TCP/IP) stack using simple HTTP requests is the concept behind HAS Single-board computers and TCP/IP-enabled microcontrollers are two examples of the types of systems that fall under this category. HAS primary objective has been to put forth a standard that would make it possible to manage, deploy, and communicate with a variety of devices. HAS does not create an effort to build the wheel; instead, it uses technologies that are already in existence and are extensively implemented, included RESTful APIs built on top of HTTP that transmit data in a JSON-encoded format.

The following novel features of HAS special are as follows;

(1)	A single and unified solution that combines all the integrated features;
(2)	Device management and provisioning.
(3)	The special firmware updated all devices using the same unique API.
(4)	Complex and intelligent rules are implemented between the actuators and sensors inside the network.
(5)	A master-slave architecture that is hierarchical and provides scalability and flexibility.
(6)	There is no need for a connection to the cloud (privacy and security reasons so, the user data are not confines of the local network).
(7)	The integrated smart phone app including plugin performs best on all top platforms, whether desktop or mobile, including Android, Windows, iOS, macOS, Linux, etc.

Section 1 represents the introduction. Section 2 describes the system architecture and design; Section 3 presents the software implementation Section 4 shows the real-life case study. Section 5 consist of discussion while Section 6 shows conclusion.

## 2. System Architecture and Design

A traditional Wi-Fi network and Ethernet are usually sufficient for a functional HAS configuration. The system uses a variety of hardware components, including Raspberry Pi 3 boards, ESP 8266, and Wi-Fi modules. In this project, the Raspberry Pi 4 model was chosen owing to the advancements made compared to the previous models. Bluetooth, etc., is unavailable on the RPi1 and 2. On the RPi, a notable feature is a row of general-purpose input/output (GPIO) pins that are available [37,38]. All current Raspberry Pi boards have a 40-pin GPIO header [33]. The RPi board can perform three functions in a HAS setup: it can serve as the primary HAS device with relay boards and sensors. It can also be the master hub for other connected devices for running Tuya (the platform of Chinese smart devices which provides cloud services for ESP8266/ESP8285-based devices). Some devices might install ESP firmware's Convert (OS), a tool that replaces the proprietary Tuya firmware with homemade firmware without stripping the device. It is vital to note that it is only compatible with Tuya-based devices.

The microcontroller is the essential component of a HAS based on the IoT. In terms of performance, the ESP 8266 Wi-Fi module comprises a collection of highly integrated wireless systems on Chip (SoCs) that provide an integrated and independent Wi-Fi network solution. The ESP8266EX version of the chip is one of the most highly integrated Wi-Fi chips available on the market. Additionally, the ESP8266EX incorporates an improved version of the L106 Diamond series 32-bit CPU from Tensilica (a firm located in Silicon Valley that works in the semiconductor industry), as well as on-chip SRAM. In addition to seventeen GPIO pins that may be assigned to different purposes by setting the proper registers, the ESP8266EX features two power pins, a reset pin, one ground wire, and two clock pins.

In most cases, the sensors and the actuators of the devices have connection of upwards are used. Observance of the firmware on devices up to date is perhaps one of the essential chores, yet it is frequently overlooked when working with many devices. To makes this work more accessible, HAS makes it possible to upgrade the firmware of devices of various sorts and models in a straightforward manner. The HAS API is a simple HTTP API that allows remote control of the main ports of the hardware (GPIO) and analog-to-digital converters (ADC). To operate programmable systems that are equipped with a TCP/IP stack, basic HTTP (Hypertext Transfer Protocol) requests must be sent to the system. Single-board computers (SBCs) and TCP/IP-enabled microcontrollers are examples of such systems. The Aspects of the application-programming interface (API) are classified into the given. Device management includes the overall status and setup of the device, port administration includes the configuration and information of ports, and reverse API calls include API calls made using reverse HTTP requests' standards, which provide a wide variety of functionality and use cases, may seem highly complicated. Although many are required for a HAS implementation, most are optional, with just a limited collection of functions required to implement it.

There are many components to the HAS ecosystem, including a HAS Server, HASOS (operating system), and additional packages and tools tailored to various settings. This application's primary component is HAS Server, which was built using Python languages. It serves as a central location that hosts a comprehensible online application. A ready-to-use operating system (OS for Raspberry Pi boards, HASOS is a server that runs the HAS Server web-based application. EspHAS is a modified firmware for ESP8266/ESP8285 devices that use the HAS API. It is available for download here. In the end, add-ons are important parts of software that extend the capabilities of the HAS Server software package. HAS devices will identify themselves, including their set-optional features and the ports exposed to other devices. Each port will define itself, including its identity, configuration, type, etc. Before moving on to the next. A complicated tree topology in a network is accomplished by integrating master-slave connections between primary devices and hubs. It is possible to handle many smart devices with relative ease. Wi-Fi or Ethernet are the modes of communication shown in Figure 2.

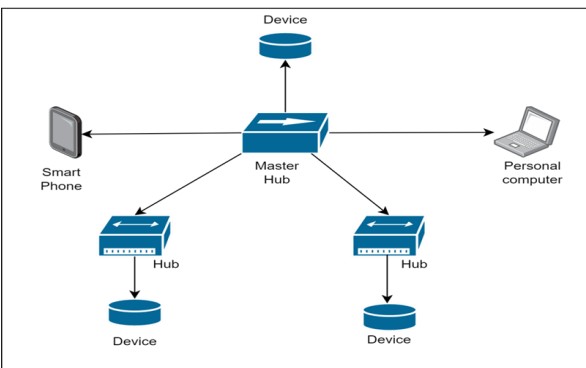

**Figure 2.** Presents the HAS topology.

It is difficult to handle a large number of devices on an individual basis in a real-world setting. As a result, a Hub will enable the administration of devices utilized by HAS to be consolidated. Hubs operate the user when communicating with connected devices, but other devices also use an API interface that enables the devices. It is possible to create devices and hubs arranged in complex hierarchies in a master-slave interaction. A device may operate master for other connected devices when using the HAS command. The master device commands the slave devices and gives them access through API calls.

A slave device may similarly operate as a master or head for additional devices. As a result, it is possible to build complicated chained master-slave setups. The master can list, add, and remove slaves using API calls exclusive to the slaves. Master(main) device recognized their slave(sub-connected) devices by using their names, while the master needed some additional information for the possibility that the weakest devices' names could be modified at any time. The detailed design is shown in Figure 3. One of three roles determines a device's access level and the most important the administrator role, have complete control over the device and can modify configuration and view; the normal role, read-write role, which can read from and write to ports can just read the values of the ports. To make automation easier, HAS enables the creation of rules that determine port values depending on various situations. Expressions may be configured either at the device or at the hub level.

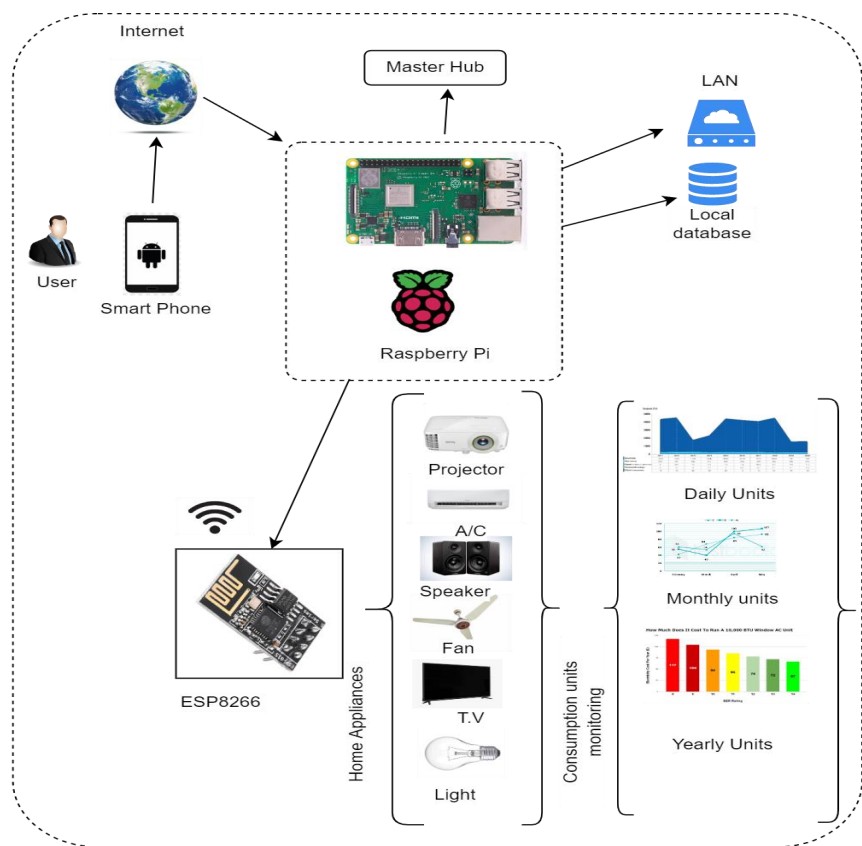

**Figure 3.** The proposed HAS full communication block diagram the current study.

The device expressions are quick, but they are only as fast as the number of ports available on the device. This will allow for the proper implementation of relationships between various devices. The HAS platform provides three notification mechanisms for users who need to be alerted about device events, such as port value change. Most of the time, HAS configurations been used in private networks, and the devices cannot access the internet. In cases when public IP addresses are accessible, port forwarding is often used to provide a solution. If port furthering is not desired or feasible, the devices

may be configured to establish a connection with a public external server and monitor for API requests. Because it enables HTTP queries to be sent to a device located inside a secure network without transmitting any port traffic, this technique is referred to as reverse HTTP. From the developers' perspective, HAS provides add-ons that are a simple and quick means of bundling alternative functions that are often related to a particular device. Add-ons may be made public or private based on the developer's demands and the licensee restrictions of the software. HAS's security is ensured by implementing the best programmable practice in today's internet-based applications. The illegal user wants to communicate with the Hub outside; HTTPS protects the communication. It guarantees that HTTP communications are encrypted, the Hub is legitimate, and the messages are not corrupted. Plain HTTP is only used locally, such as in buildings, etc., to communicate among the Hubs and the devices under its supervision. A TLS certificate helps achieve the security objectives listed above when combined with HTTPS. Encrypt is responsible for generating and renewing the TLS certificates. When a certificate's expiration date approaches, this operation is initiated automatically on the Hub. HAS is used to access the Hub remotely (for administrative purposes).

The HAS protocol uses ECDSA (or a matching algorithm) to encrypt the login information authentically. The online login, the administrator password, and user name established on the Hub may also be utilized. The three rules of the API determine the rights granted to API requests: regular user, view-only user, and administrator. Each role has its own set of permissions. API queries give authentication data via the JSON Web Token (JWT) specified by RFC 7519. Using a shared secret (also known as a password) ensures the caller's legitimacy. Instead, we may have utilized HTTP Digest, HTTP Basic Authentication or a session management method that relies on cookies and combines them with a conventional login form. While Basic authentication is vulnerable when delivered over unencrypted lines, Digest authentication is overly complex and necessitates the exchange of numerous messages, which is unnecessary. The cookie/session-based technique is susceptible to session theft attacks and may be unsafe when sent over unencrypted networks. If the vulnerability is detected, the inbuilt Over-the-Air (OTA) mechanism (firmware update) guarantees that the Hub and its linked devices are always running the most recent available version, enabling us to deploy security immediately It shows that the user and admin use the proposed HAS have the application installed in their smartphone and connected with internet services with the master hub. The master hub includes the communication devices such as the Raspberry Pi module with a local area network (LAN) and local database. The local database saves user records, login information, and dashboard data. Further, the master hub communicates with a Wi-Fi device (ESP8266) to control and monitor the home appliances such as fan, A/C, TV, projector, etc., of all the rooms, including living rooms, bedroom, and dining room. HAS also calculated the consumption units of daily, monthly, and yearly based on the home appliances. The proposed communication diagram of HAS is shown in Figure 3.

## 3. Software Implementation

Google's Android Studio delivers the quickest tools available for developing applications for any kind of Android device. It allows concentrating on developing one-of-a-kind, and high-quality applications since having access to world-class performance tools, debugging, code editing, a customizable build system, and an immediate build/deploy system. Android Studio is the official Integrated Development Environment (IDE) for developing Android applications with its foundation in IntelliJ IDEA. The HAS application has been developed in android studio 4.2 versions, including JDK (java development kit) 8.1 with a linear layout.

### 3.1. Programming Languages (Java, Python)

Java languages were developed by Sun Microsystems and initially published in 1995. There are various sorts of devices, including Java, as a mobile phone application, and

mainframe computers. Java does not convert to native processor code; instead, it depends on a "virtual machine" that recognizes a transitional format known as Java byte code. Every platform, which runs Java, requires a virtual machine (VM) implementation. Dalvik is the name given to the first virtual machine on Android. They are responsible for translating bytecode, a collection of instructions comparable to the assembly language found in CPUs, into executable code that can be executed on a computer's processor. Python, Ruby, Objective, C, and Java++ are all included in the Raspberry Pi's default software installation. Moreover, the letter "Pi" in Raspberry Pi is derived from the programming language Python, and the concept of programming is included inside the gadget's name. Python is a high-level, general-purpose, interpreted, and dynamic programming language that is extensively used today. In addition, the language's design philosophy prioritizes the readability of code, and its syntax enables programmers to express ideas in fewer code lines than feasible in other programming languages like JAVA or C++. The language has features designed to allow for clear programs on a small and big scale.

### 3.2. Configuring the Web App

HAS server provided an easy interface for users and administrators, delivered with a progressive web application (PWA) platform. It is intended for mobile devices, including Android phones, laptops, tablets, and desktop computers.

First, the program should be downloaded and installed, and since it is a Progressive Web App, it should be put on the home screen. It is possible to remove the HAS app at any time after it has been installed since it will appear in the device's programs list once it has been loaded. A demo interface is available to directly access the server when a first-time user logs in (as shown in Figure 4. However, for security concerns, it is strongly advised that put a strong password on the app's settings page before using it. When utilizing HAS Server, the dashboard is the interface which most probably used by the user. They may use this screen to build sets of panels. The user may conduct different operations, such as adding, moving, removing, resizing, and configuring widgets, among others, in the edit panel mode. Widgets often need the selection of one or more ports. Upon interaction with the widget, the values of the ports will be shown and updated.

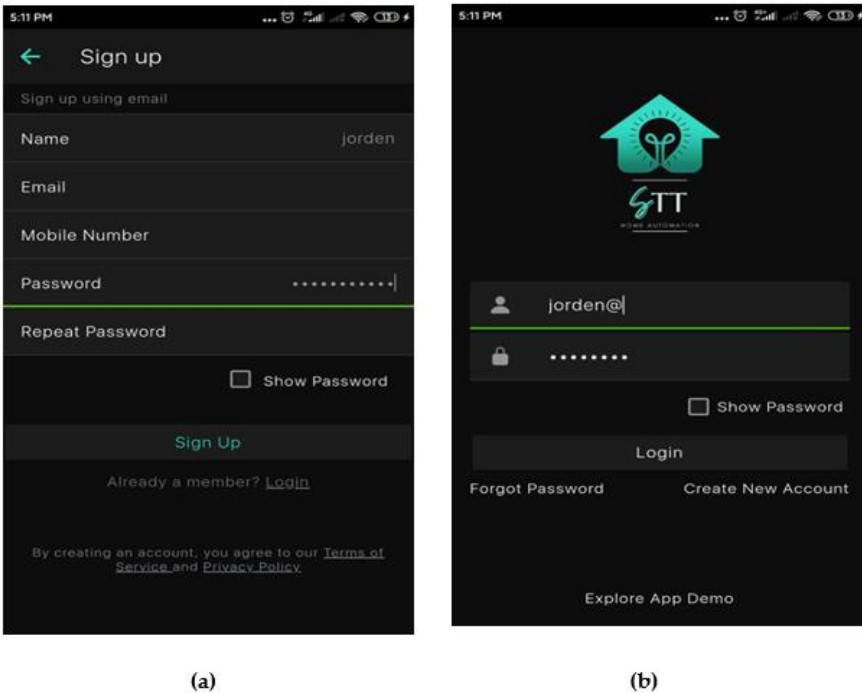

**Figure 4.** Present the user profile on HAS server application. (**a**) Signs up at first-time, putting password and email. (**b**) Login to the HAS server app.

## 4. Real Home Automation Case Study

The section represents the usability of the HAS home automation app in a real-world setting. The setup comprises a two-story home with four rooms and two bathrooms. It also has a kitchen and backyard. In this situation, the home automation app is utilized for various functions, including controlling all the appliances, such as television, set-top box, air conditioner, projector, etc. Users can create a detailed and personalized entertainment Program Guide to keep track of everything they watch on the television. Make use of Routines and Scenes to schedule all of the appliances. Create Workflow to perform a series of operations in response to changes in room temperature, motion, etc. The appliances' real-time power consumption and energy use statistics Like Google Assistant and Amazon Alexa control all of the appliances with their voice.

### 4.1. Controlling Home appliances (A/C, Television, Speakers, Ceiling Lights)

The home automation app may be used to operate the A/C, allowing it to be turned on and off without needing a remote. This process may be accomplished by using a smart plug for the air conditioning equipment. A feature that can be added quickly should need to arise, which looks very similar to a thermostat, is the ability to control degrees.

Furthermore, smart lighting has the potential to make our daily lives simpler. Another benefit is the reduction of energy use. As shown in Figure 5.

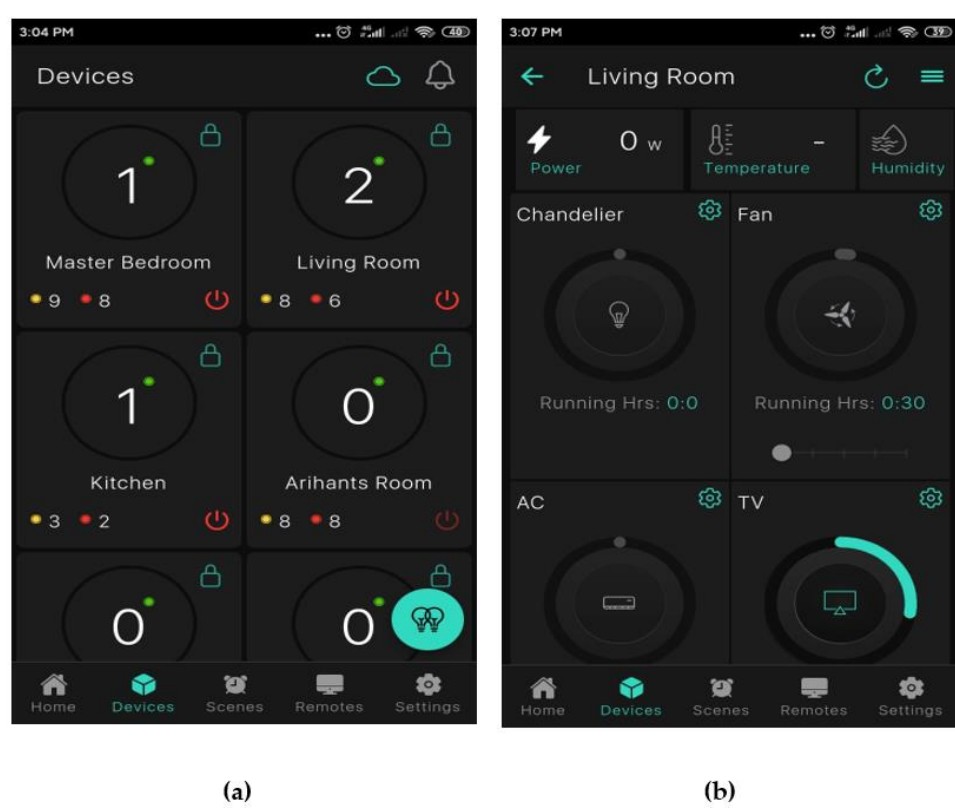

|         |         |
|:-------:|:-------:|
| **(a)** | **(b)** |

**Figure 5.** The dashboard of the controlling all appliances in the rooms. (**a**) Switching on-off all the devices of the rooms. (**b**) Turn on and off specific room's appliances.

Large residences with several rooms may waste a significant amount of energy merely by keeping the lights on in areas with no need for appliances. At the same time, most people turn on the lights in some places in the home before leaving home or to sleep because they forget to turn off the home appliances in such situations, it is necessary to check the lights and turn them off by using the smartphone's mobile application. Moreover, a smart lighting management system provides security to the home by increasing the amount of protection available and controlling speakers and television, as shown in Figure 6.

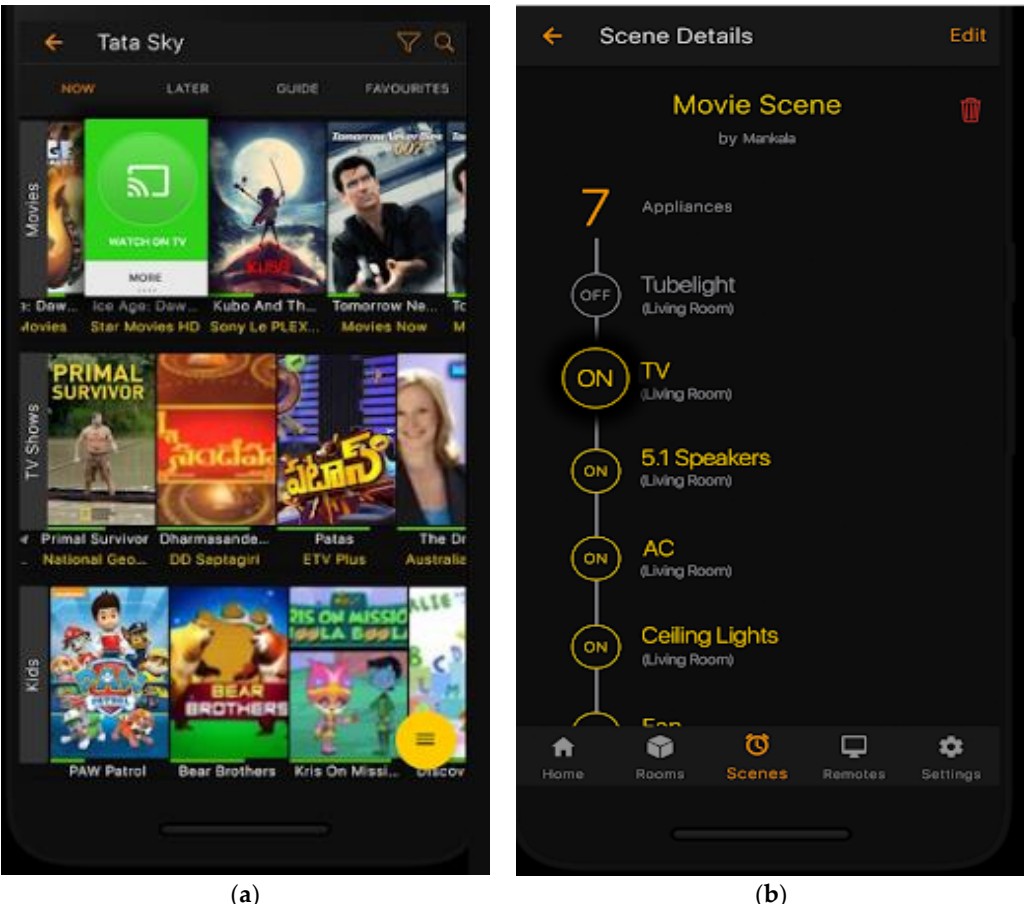

**Figure 6.** The dashboard of the movie scenes and other appliance controlling. (**a**) Presents TV appliances controlled by on-off switch on dashboard. (**b**) Movie scene appliances along with A/C and speaker controlling switches.

### 4.2. Energy and Power Monitoring

Recently, the entire world has been trying to use less energy to make the globe greener. So, for the renewable energy sources, solar energy system can be used as an optimal solution of energy efficiency.

Photovoltaic panels have efficiently converted the energy from the sun's beams into electrical power, which is now less valuable than before. It is possible to enhance the benefits of renewable energy in a home by integrating the energy savings provided by solar systems with advanced technologies. Using solar electricity, it is possible to automate home solutions. Moreover, adopting solar power-based home solutions may lower energy costs while increasing energy efficiency. Furthermore, the solar power-based smart home solutions can decrease individual carbon footprints, reduce total environmental impact and emit zero emissions. In the case study of the automated system, in the absence of sensors, the user can set a list by activating the button to enable the schedule. All of the modifications may be adjusted on the app's final four screens, titled Morning and Evening Time and factor, respectively. The user can decide the time of day when the irrigation should begin and the volume of water applied by modifying the Morning/Evening factor. This app predicts the daily home power consumption in real-time of the master room to the kitchen room, as shown in Figure 7.

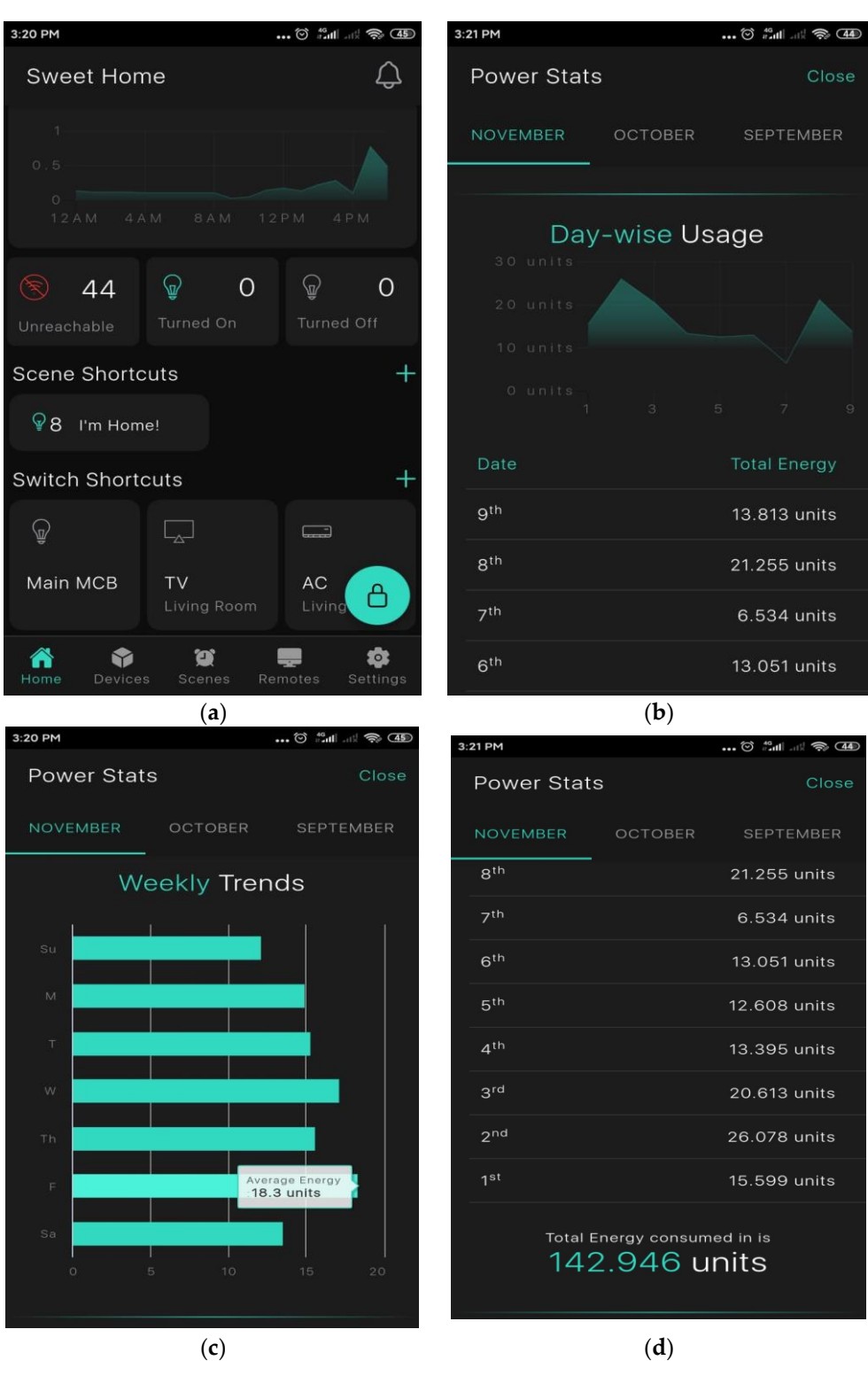

**Figure 7.** Presents The dashboard of the home appliance power consumption. (**a**) Presents total consumption of home appliances power. (**b**) Daily wise power consumption of all the devices. (**c**) Weekly wise power consumption of all the appliances. (**d**) Monthly wise power consumption of all the appliances.

### 4.3. Controlling Home Appliances by Using Google Assistant (Voice)

The home automation app controls the home appliances using voice, allowing it to be turned on and off without needing a remote. This process may be accomplished by employing Google Assistant and Amazon Alexa in this case study, as shown in Figure 8.

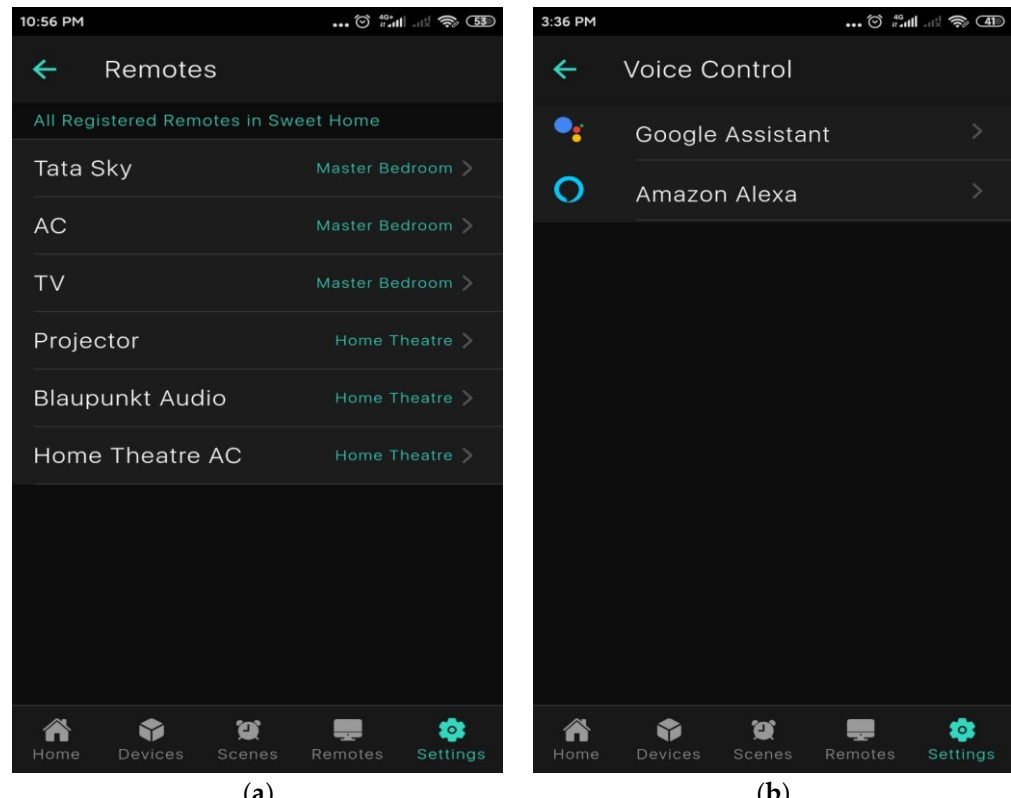

**Figure 8.** Presents the remote and voice control setting. (**a**) available the remote (TV, AC, Projector,) setting commands. (**b**) include voice control (Google Assistant, Amazon) setting option.

Table 3 shows a list of the sensors and actuators utilized in the case study and their major features for convenience. It is understood that any kind of sensor or actuator may be employed in the HAS home automation app architecture.

**Table 3.** Sensors/actuators implemented by HAS for the given real case study.

| Actuator/Sensor | Measurement Range | Name | Consumption | Expense |
|---|---|---|---|---|
| Power meter | 90–250 V | Mai-90 | 2 W | $20–30 |
| Temperature | 5–50 °C | ME81H | 400 mA | $15–25 |
| Touch sensor | logic | Sonoff Touch | 0.6 W | $20–25 |
| 4-relay module | - | SunSmart | 5–70 mA | $4–6 |

## 5. Discussion

This research proposed a solution for smart home automation system tasks using Raspberry Pi boards and ESP8266 chips. Both options are cheap and are simple to manipulate. In addition, the HAS system implements a fundamental core API, which allows a more adaptable network architecture. HAS was designed to become a fully working intelligent home prototype. It includes many features, such as control, automation, security, and monitoring. This system can be further improved and developed. One of the contributions of this paper is conducting a review of current (within the last ten years) studies that have been published in the academic literature, as well as commercial solutions for home automation systems as described in Tables 1 and 2. This paper shows a detailed hardware and software implementation solution compared to other literature research.

Most of literature [4–34] about HAS have been constructed with minor functionalities and implemented different communication, technologies, user interfaces, controllers etc., as presented in Table 1 provide firmware open source, which means no need for a third-party

hub; all devices communicate in the same language API and are operated in the same manner. The supported and manage devices have undergone meticulous testing, and the installation process has been well documented. This does not imply that HAS cannot have other devices added to it; there are add-ons available that give bridges and adaption layers to various technologies, networks, peripherals. In the HAS architecture, one device can work as a master the other performed slave, while one can work slave and the other device work as a master simultaneously. Thus, it makes it possible for a reasonably high number of incoming requests to be sent to each device per second, increasing the system's scalability. In this article, we provide a genuine case study, which takes place in a real house, as well as all of the features that the proposed system, with the application, offers to make living simpler and more affordable.

The solution can be effectively utilized in practice. This article explains how the system is constructed, how the application may be configured and installed, what functions are included, and which tools and resources may be used to have a smart home. HAS provided flexibility and simplicity. Furthermore, one of our goals was to provide an affordable smart home. Home automation maintenance and deployment include high costs. This cost will increase even more components that make up the system and the kind of technologies used increase.

Our approaches motivate by the controlling of the firmware of each device. Controlling the firmware enables the full power of each device to be leveraged, the device is customized to meet the requirements of each user, and critical security updates are provided, all while unifying the API. Because we prioritize users' privacy, HAS ensures that user information is never shared or outside of the building (i.e., the home local network). Users who do not have a technical background are included in HAS target audience. These users prefer user-friendly applications. This is likely one of the most significant differences between HAS and its competitors.

Furthermore, perhaps most importantly, the fact that we have provided a solution that is applicable and effective in the real world should be emphasized. The proliferation of sensors built into consumer electronics in the not-too-distant future will make it possible to automate virtually every facet of our daily lives at home. Monitoring the relative humidity of the air is going to be a feature coming soon to HAS. Mold can be caused by high humidity levels, which can also result in financial damage. A low-level humidity produces various respiratory illnesses and also increase cancer rate and different diseases [58–62] and makes it easier for viruses to proliferate, while a high humidity level leads to condensation and mold growth.

Consequently, monitoring humidity levels by a humidity sensor will defend both buildings and the items inside it.

Furthermore, the proposed systems reduce energy consumption by 12% compared to the researcher's literature [63]. Because the proposed system uses minimal electronic equipment, and the devices are connected using API. There is no need for Hub. These benefits will create a positive impact on the environment as well as on consumers' bills.

Additionally, it produces the alert to the customer if the relative humidity of the indoor environment fluctuates to undesirable levels. Video surveillance integration into HAS will be yet another project for the near future.

## 6. Conclusions

This paper described various interconnection systems and data sources such as sensors, and actuators to enable multiple home automation implementations. The system is known as HAS (Home automation system). It operates by connecting the efficient API, which is the key to a straight forward and universal communication method. HAS often used the actuators and sensors with have upwards communication and implemented the HAS (API). Furthermore, most of the devices of the HAS (home automation system) used RPi boards and ESP8285 chips. A smartphone web-based app has been developed that enable users to full control of a wide range of smartphone-connected devices. The application

is user-friendly, adaptable, and beneficial for consumers and disabled people. It has the potential to be further extended via the use of various devices. The main objectives of this work are to make our home automation system more secure and intelligent. HAS is computationally efficient and effective. The future study of HAS will present more features such as video monitoring, which is for single-board computers, to allow a user to operate to video cameras inside the browser.

**Author Contributions:** Conceptualization, M.A.K. and I.A.; methodology, M.A.K.; software, I.A.; validation, A.N.N. and A.E.-S.A.; formal analysis, Y.I.D.; investigation, I.A., S.R. and H.M.; resources, I.A. and Y.I.D.; data curation, I.A.; writing—original draft preparation, M.A.K., I.A., H.M. and S.R.; writing—review and editing, Y.I.D.; visualization, A.E.-S.A. and I.A.; supervision, A.N.N. and M.S.; project administration, A.N.N., E.T.E. and M.S.; funding acquisition, E.T.E. All authors have read and agreed to the published version of the manuscript.

**Funding:** This research was funded by Taif University Researchers supporting Project number (TURSP-2020/159), Taif University—Saudi Arabia.

**Institutional Review Board Statement:** Not applicable.

**Informed Consent Statement:** Not applicable.

**Data Availability Statement:** The datasets used in this investigation are available on request from the corresponding author.

**Acknowledgments:** The authors would like to thank Taif University Researchers supporting Project number (TURSP-2020/159), Taif University—Saudi Arabia.

**Conflicts of Interest:** Authors declare no conflict of interest.

### Abbreviations

| | |
|---|---|
| IOT | Internet of Things |
| API | Application Programme Interface |
| HTTP | Hypertext Transfer Protocol |
| REST | Representational state transfer |
| JSON | JavaScript Object Notation |
| HAS | Home Automation System |
| PIC | Microchip Technology in Chandler |
| MQTT | Message Queuing Telemetry Transport |

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
