# Peer review of "Smart Android Based Home Automation System Using Internet of Things (IoT)"

_sustainability, doi:10.3390/su141710717_

Round 1
Reviewer 1 Report
The introduction section should introduce the aims of the research.
The researcher didn’t provide a clear methodology for the research. What objections were set and how to achieve these objectives. No clear formation of the central research question that is linked with the methodology. I am not sure if this is a case study based research or experimental design?
How this research work is related to sustainability is not clear.
Just critical writing or work.
The researchers Present a detailed design of the proposed work in figure 3, however unable to explain what's new in this. It's a general concept diagram and already in place everywhere, what's new in this?
Researchers claimed that A smartphone app (HAS) has been developed that allows users to control a wide range of home appliances and sensors from their smartphones. What is new in this app? How it is different from other apps already used. I believe these apps like Alexa, google assistant are more widely used and provide more compatibility.
Based on the above point my decision is to reject the research article.
Author Response
Dear reviewers,
Thank you for comments concerning on our manuscript entitled “Smart Android Based Home Automation System Using Internet of Things (IoT)” sustainability-1753955. Those comments are all valuable and very helpful for revising and improving our paper, as well as the importance guiding significance to our researches.
We have substantially revised our manuscript after reading the comments provided you, and revised portion are marked in red in the paper.

Reviewer 2 Report
This paper presents an interesting case study on home automation systems. The structure of the work responds to what is expected of a scientific article. The wording is correct and appropriate and allows understanding of abstract aspects of engineering. The authors demonstrate extensive knowledge of the state of the art, propose a methodologically correct case study, and, as a strong point of their work, propose an extensively analyzed applied model. The references used are pertinent, excellent and are correctly referenced.
For all these reasons, I consider that the article is of interest from a scientific point of view and also from a social point of view and scientific dissemination. In my opinion the work can be published in its current state.
Author Response

(The authors gave the same response as above.)

Reviewer 3 Report
The authors have developed a home automation system using Internet of Things. But a few major concerns need to be considered.
1. The authors need to check the vocabulary thoroughly in the entire manuscript.
For instance, the sentence in line 111-112 are confusing, the authors should make a clarification and maintain the same notation either HAS or HIS in the whole manuscript.
2. I would suggest the author to have the following reference for the IoT technologies, where they clearly represented the Internet of Things applications involving smart home automation system.
Sukhavasi, S.B.; Sukhavasi, S.B.; Elleithy, K.; Abuzneid, S.; Elleithy, A. CMOS Image Sensors in Surveillance System Applications. Sensors 2021, 21, 488. https://doi.org/10.3390/s21020488
3. All figures should follow the same font and notation, the authors need to makesure all images are following the format. For instance, from figure 7 to figure 9, the images are not clearly represented with labels. So, every figure inserted in the manuscript must be clearly represtened with label and the corresponding label description should be provided.
4. Some abbrevations are missing in the table provided, please check the manuscript thoroughly to find the ones and add them in the table.
5. There are grammatical typos in the manuscript including the title where the authors wrote "Internet of Thing" which is supposed to Internet of Things., so, please change accordingly.
Author Response

(The authors gave the same response as above.)

Round 2
Reviewer 1 Report
I believe the authors have invested a lot of efforts to improve the manuscript and responding to my queries. I believe that with minor language editing the article is acceptable in its current form.
Author Response
Dear Reviewer,
Thank you for your comments concerning on our manuscript entitled “Smart Android Based Home Automation System Using Internet of Things (IoT)” sustainability-1753955. Those comments are all valuable and very helpful for revising and improving our paper, as well as the important guiding significance to our research.
We have substantially revised our manuscript after reading the comments provided by the reviewers, and the revised portion are marked in red in the paper.

Reviewer 3 Report
The authors have addressed the major concerns which were highlighted earlier. However, by verifying the revised manuscript, there are a few concerns regarding the organization of the manuscript, which needs to be done for better readability and interest.
1. Figure description has changed for Figure 1, please fix it.
2. Figure 4,5,6 are merged with bottom of the pages and are not clear, needs to be replaced with a clear one. Please make sure all the figure descriptions should come under the figure in the same page.
3. Figure 7 is completed merged itself with all the figures, It need to be seperated according the description labels a,b,c and d. Please make sure all the figure descriptions should come under the figure in the same page.
4. A table showed up in page 19, with no table number is alloted to it. If you want to keep it, you need to provide a table number to it and refer it in the paragraph.
5. I would suggest the authors to make sure the manuscript is having proper figure representations and table representations.
Author Response

(The authors gave the same response as above.)
